# Drowning in the Information Flood: Machine-Learning-Based Relevance Classification of Flood-Related Tweets for Disaster Management

Eike Blomeier [1], Sebastian Schmidt [1,*] and Bernd Resch [1,2]

1   Department of Geoinformatics, University of Salzburg, 5020 Salzburg, Austria;
    eike.blomeier@stud.plus.ac.at (E.B.); bernd.resch@plus.ac.at (B.R.)
2   Center for Geographic Analysis, Harvard University, Cambridge, MA 02138, USA
*   Correspondence: sebastian.schmidt@plus.ac.at

**Abstract:** In the early stages of a disaster caused by a natural hazard (e.g., flood), the amount of available and useful information is low. To fill this informational gap, emergency responders are increasingly using data from geo-social media to gain insights from eyewitnesses to build a better understanding of the situation and design effective responses. However, filtering relevant content for this purpose poses a challenge. This work thus presents a comparison of different machine learning models (Naïve Bayes, Random Forest, Support Vector Machine, Convolutional Neural Networks, BERT) for semantic relevance classification of flood-related, German-language Tweets. For this, we relied on a four-category training data set created with the help of experts from human aid organisations. We identified fine-tuned BERT as the most suitable model, averaging a precision of 71% with most of the misclassifications occurring across similar classes. We thus demonstrate that our methodology helps in identifying relevant information for more efficient disaster management.

**Keywords:** disaster management; relevance classification; social media; semantic analysis; BERT





## 1. Introduction

Influenced by increasing urbanisation and the growing impact of human-made climate change, disasters caused by natural hazards are becoming increasingly frequent. The timely and reliable assessment of such events is therefore gaining importance. In central Europe, major flood events are the main cause of large-scale damage and loss of life [1]. Traditionally, remote sensing data, which can come from satellites or airborne sensors (e.g., mounted on drones), are used to observe and delimit a flood. However, this use has temporal (e.g., satellite repetition rates), technical (e.g., cloud coverage) and financial limitations, which makes it of interest to aid organisations to also use additional data sources. Since the amount of posts on social media platforms rises significantly during a disaster [2–5], such data have been employed for this purpose for years.

Before, during and after disasters, the use of social media is above average [6]. This concerns not only the pure number of posts, but also applies to specific tools like Facebook Safety Check or the Google Person Finder, through which people can now request the safety status of their friends. While these functions might provide important information for individuals, they do not help emergency responders to obtain a bigger picture of the situation. This input is crucial, though, since in many emergency situations, no or insufficient information is available shortly before or until emergency services arrive [7]. By enhancing the situational awareness and accelerating the spread of information, the spatiotemporal distribution of disaster-related social media posts therefore brings major benefits to the assessment of disaster damage [8,9] and to situational awareness. A central question here, however, is which content from social media is actually relevant for use in the disaster management process, and how it can be filtered and categorised.

For this, the concept of "relevance" should first be addressed, as no unanimous definition exists. One definition in [10] describes relevance as a measure of the effectiveness of the contact between a source and a destination in a communication process. Ref. [11] considers relevance as a multidimensional, systematic, user-centric, measurable concept influenced by both cognitive (internal) and situational (external) elements. Ref. [12] calls relevance the central concept of information retrieval theory, stating that observations are relevant only if they are members of a minimal stored set from which an answer to the respective question can be inferred. Ref. [13] describes relevance as the "correspondence in context between an [information] requirement statement and an article, i.e., the extent to which the article covers material that is appropriate to the requirement statement". Ref. [14] determines that information is only relevant if it can provide necessary input for a user, which was the definition we followed. Other definitions split relevance into adequacy and usefulness [15,16]. To clearly clarify which content is relevant to the disaster management process, we worked closely with first responders, which is a unique feature of our paper. Based on an iterative process, we created a training data set that assigns a relevance category to each tweet and it was used to train our models.

The short-message service Twitter (now X) is particularly suitable for collecting data in crisis situations. Until June 2023, Twitter allowed a free, representative sample of Tweets to be accessed via various Application Programming Interface (API) endpoints. One possibility is to extract explicitly georeferenced Tweets, most of which become geocodable via a "place" tag set by the user for the respective Tweet. Numerous studies have already shown that significant statements on geo-social phenomena can be derived from this selection [17–21]. A major difficulty, however, is reducing the large, rather unstructured amount of data for each use case, i.e., to consider only relevant Tweets. Traditionally, this was achieved through keyword-based filtering. For this, a list of relevant keywords for an event had to be created in advance, which were then searched for in the text corpus. Due to linguistic and grammatical diversity, however, this leads to blurring in the filtering, e.g., with regard to semantically ambiguous words.

To overcome this limitation, we investigated various machine learning approaches for relevance classification of flood-related Tweets. We compared the performance of more traditional, rather established models—Naïve Bayes (NB), Random Forest (RF) and Support Vector Machine (SVM)—to two more advanced models: a Convolutional Neural Network (CNN) and a fine-tuned Bidirectional Encoder Representations from Transformers (BERT) model. We chose these models because they are particularly frequently used approaches in the field of Natural Language Processing (NLP) and social media analysis. This work was motivated primarily by the Ahr Valley flood of 2021, in which the added value of social media data from relief organisations was confirmed. For this reason, the focus of the method development was on German language content, for which there was also a research gap. Consequently, we aimed to address the following research question:

Which machine-learning-based approach is best suited for relevance classification of flood-related, German-language Tweets?

## 2. Related Work

Data from social networks have already been used in many studies relating to disasters. In particular, topic modelling methods have been employed. Ref. [17] use Latent Dirichlet Allocation (LDA) to extract such topics from Tweets for real-time monitoring of disasters. Advanced algorithms for semantic classification, such as a CNN [22], BERT [23,24], and a Graph Neural Network (GNN) [25], have also been proposed. Some approaches go beyond purely textual analyses and also include the content of images [26]. However, these studies generally do not consider the relevance of posts for use in disaster management.

To convert the previously mentioned definitions of relevance into concrete criteria is challenging. For an extensive overview of potential abstract (e.g., necessity, topicality, impersonality) and factual criteria (e.g., referenced geographic locations, currency), see [7]. In their paper, Ref. [27] classify Tweets into three categories: *off-topic*, *on-topic and relevant*

and *on-topic but irrelevant*. They first perform keyword filtering and then assign the classes manually, also considering imagery content. In their study, a post is deemed relevant if it can enhance situational awareness. A similar distinction is also made in [28], who differentiates posts that are relevant or irrelevant of situational awareness. Ref. [29] also consider retweet behaviour as indicative for semantic relevance, reasoning that frequently reposted information has a stronger connection to an event.

Several machine-learning-based approaches have been proposed for relevance classification of social media content. For example, Ref. [30] use a CNN as a first step to classify Tweets as informative or uninformative. In a second step, they sort the informative messages into eight categories of actionability using a radial basis function SVM. Ref. [31] compare the performance of several machine learning algorithms on a supervised multi-class classification problem. They conclude that the linear SVM outperforms the other algorithms (including NB and RF) in six of the seven classes, using a combination of unigrams and bigrams. Ref. [32] also propose an SVM-based methodology. Ref. [33] investigates stacking a CNN and an Artificial Neural Network (ANN) to classify Tweets on Hurricane Harvey as informative or non-informative. Ref. [34] compare k-nearest neighbour (kNN), multinomial NB and SVM, amongst others. Ref. [35] uses a BERT model to classify Tweets as relevant and irrelevant for the 2020 Jakarta flood, adding dense and dropout layers to the pre-trained model for fine-tuning. A Cross-Attention Multi-Modal (CAMM) deep neural network is proposed in [36] which combines information from text and imagery. However, they only distinguish *informative* and *non-informative* classes. Ref. [37] utilise BERT and XLNet to sort Tweets on Hurricane Harvey into relevance categories, showing that pretrained language models outperform traditional methods such as SVM. Ref. [38] use a fine-tuned RoBERTa for the classification of disaster-related Tweets, as well as a vision transformer model for attached imagery, achieving an accuracy of up to 98%. In their paper, Ref. [39] suggest using a GNN to combine textual information, imagery content and time for flood-related Tweets.

Most of these approaches refer to English-language Tweets. In general, there is a large bias towards English in NLP applications. This can be problematic if a language is morphologically more complex, e.g., German [40]. Accordingly, some models developed based on English training data do not necessarily work well for other languages or require the translation of such Tweets, which can lead to information loss and is costly. Furthermore, the suitability of BERT-based models, which represent a big step forward in NLP, has hardly been considered in relevance classification. Furthermore, a more precise categorisation of relevance than a binary classification is rare. Our paper addresses these research gaps.

## 3. Materials and Methods

In our research, we address a multi-class supervised classification problem [41], where a single class has to be predicted for each input Tweet. This is also referred to as "one-class" classification [42]. Figure 1 shows a schematic overview of our workflow.

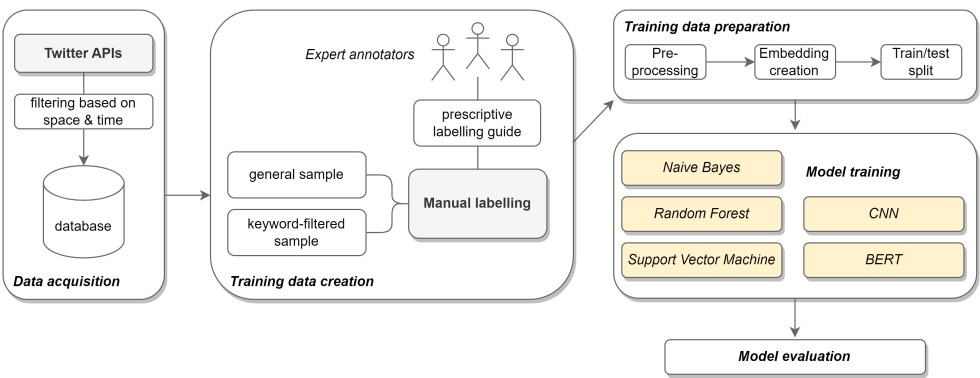

**Figure 1.** Workflow for comparison of models for relevance classification of flood-related Tweets.

### 3.1. Data Collection and Labelling

For the creation of our training data set, we only used geo-referenced Tweets that were sent within a bounding box around Germany in the year 2021. We extracted Tweets for the entire year, since otherwise our algorithms might have been biased towards the Ahr Valley flood in July, one of the most devastating disasters in recent German history. We retrieved Tweets using both the REST and the streaming API of Twitter, via which georeferenced data can be accessed. In our data collection approach, we followed [18,43].

To create training data for our machine learning models, we manually annotated Tweets. For this, we randomly extracted 10,000 Tweets from our data set without any further specifications. To increase the proportion of potentially relevant Tweets, we additionally performed keyword-based filtering (cf. Table A1). Accordingly, two thirds of our training data set contained at least one of these keywords, while the remaining training data corresponded to a random sample of all posts sent in the study area and within the specified period. To obtain more congruent results, we decided to use a prescriptive annotation approach, for which a labelling guide was created [44]. Our labelling process was carried out in collaboration with experts from German and Austrian disaster management organisations (Bavarian Red Cross, Austrian Red Cross, Federal Agency for Technical Relief). The annotators received only the text for each Tweet and no further information; i.e., it was assumed that each Tweet was spatially and temporally relevant. To filter out unusable Tweets, two additional categories were introduced for posts in other languages and without proper text (cf. Table 1). We decided against a continuous rating (e.g., relevance percentage), as this would be more difficult to label consistently. Instead, we created a more generalisable four-category scale between "very relevant" and "not relevant". In order to delimit the categories, we provided some examples for each category in our labelling guide (cf. Table A2). Each Tweet was then labelled by three people. The result was only appended to our training data set if an inter-annotator agreement of at least 2/3 was satisfied. All Tweets labelled as "not in German language" or "no text contained" were discarded. After labelling, 4634 Tweets remained. The distribution of classes can be seen in Figure 2.

In the next step, we undersampled our data set, since training a robust model using imbalanced data is difficult [45]. For this, we randomly selected 178 Tweets from each class, i.e., the number of Tweets in the smallest category, to achieve identical class sizes. The resulting data set was then split once into a stratified training (75%) and testing (25%) subsets. Then, we used the pre-trained GBERT$_{base}$ model to tokenise the Tweets and create embeddings, i.e., to convert them into vector representations the machine learning models can use.

In total, we set up five different machine learning algorithms for comparison purposes, which will be briefly explained in the following sub-sections.

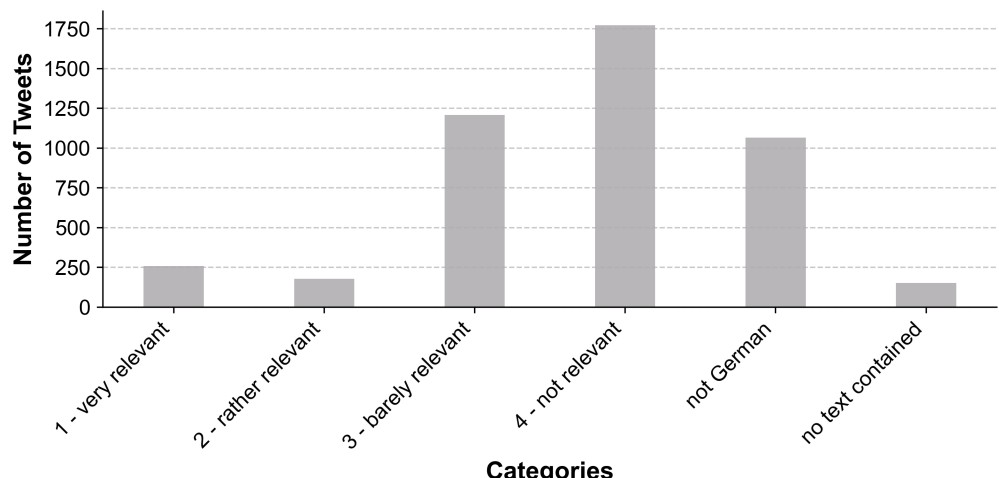

**Figure 2.** Distribution of categories for training data.

**Table 1.** Labelling categories. See Table A2 for concrete examples.

| Category | Explanation |
|---|---|
| 1—very relevant | A Tweet that is very helpful in supporting crisis management in case of a flood (e.g., Tweets referring to destructions, critical infrastructure). |
| 2—rather relevant | A Tweet that is somewhat helpful in supporting crisis management in case of a flood (e.g., Tweets mentioning efforts by first aid organisations, people that are not affected). |
| 3—barely relevant | A Tweet that is not really relevant but refers to a flood event (e.g., declarations of solidarity, appeals for donations, political or religious statements). |
| 4—not relevant not German | A Tweet that has no relation to a flood event. A Tweet that is not written in German language. |
| no text contained | A Tweet that contains no text, e.g., only emojis, links or user handles. |

### 3.2. Naïve Bayes

The NB classifier is a probabilistic machine learning model, which creates a separate model for each possible category, and thus is a generative model [46]. The core of NB is the Bayes' Theorem:

$$P(A|B) = \frac{P(B|A)P(A)}{P(B)} \tag{1}$$

where the probability *P* of an event *A* happening is computed under the assumption that another event *B* already has occurred independently. This assumption of conditional independence, however, is rarely matched in real-life cases, which is why NB tends to compute overconfident probabilities that are often very close to 0 or 1 [46]. Another common issue is the "zero-frequency problem"; i.e., the model has to classify a parameter which is not represented by a class-attribute combination. Following the commutative, distributive and associative properties of multiplication, the output likelihood in this case is always zero [47]. Nevertheless, NB is widely used in NLP problems like sentiment analysis, spam filtering or recommendation systems [46]. We employed the *ComplementNB* classifier from *sklearn.naive_bayes*, which uses the statistics from the complement of each class to compute the model's weights. Hence, it generates more stable parameters and regularly outperforms multinomial NB [48].

### 3.3. Support Vector Machine

An SVM can be used to divide a multi-dimensional space. For this, a hyperplane is fit to separate an input data set in any given dimension into two clusters [49]. Since there is theoretically an infinite number of hyperplanes, an SVM tries to identify the hyperplane that maximises the margin between the classes. This is accomplished by finding the maximum-margin hyperplane, i.e., selecting the most similar examples which have different class labels (i.e., support vectors) to draw the hyperplane orthogonal to the connecting vector [46,50]. For this, an SVM uses kernel functions to project the input data into a higher-dimensional space where clusters can be linearly separated. This allows the SVM to identify a feature space with only the necessary dimensions to separate the input data, thereby avoiding the so-called curse of dimensionality [49,50]. Furthermore, a soft margin is added to the hyperplane, since noise will make linear separation impossible for real-world examples [49]. It allows some data points to fall to the "wrong" side of the margin. However, setting the parameters for this soft margin is complicated, as it requires a trade-off between avoiding overfitting on the training data and achieving appropriate generalisation [50].

### 3.4. Random Forest

RF uses a combination of different decision trees to solve regression or classification problems [51]. The cost function of a decision tree tries to maximise the information gain at each node, thereby minimising the entropy, i.e., the uncertainty of a random variable. As a result, decision trees often grow very large and may overfit the training data [46,52]. RF overcomes the limitations of a single decision tree by combining a large number of trees operating as a committee, leading to a substantial improvement in accuracy and accordingly outperforming any of the constituent models [53]. Bagging (bootstrap aggregating, i.e., sampled training sets) and boosting (i.e., weighted training sets) can be used to ensure that the constituent decision trees are distinct [46]. We used the *RandomForestClassifier* from *sklearn*. To find the best hyperparameters for the RF classifier, we performed a randomised search and a subsequent grid search. Our optimal RF had 33 seeds, a maximum depth of 210, a min_samples_split of 21 and a min_samples_leaf of 3.

### 3.5. Convolutional Neural Networks

The architecture of a CNN is a subcategory of ANNs, where typically only the last layers are fully connected, while the other hidden layers are only connected to corresponding parts of the preceding layer [54]. As data pass through the depth of a CNN, the input vectors are reduced until the output vector reaches a specified size, e.g., $1 \times 1 \times n$ [55]. At the core of a CNN is the convolutional layer, where learnable kernels are linked to local regions of the input data. As the kernel slides across the input vector, it performs scalar product computations, creating an activation map, which is then propagated to the subsequent layer. Subsequent non-linear computations, often achieved using the rectified linear unit (ReLU) activation function, help eliminate irrelevant information. In the pooling layers, the input feature map is downsampled to reduce the computational complexity. By sliding across the feature map, these layers merge values within defined regions, producing a single value. One of the most common pooling functions for this is max pooling, which retains only the maximum value within each region. In fully connected layers, each neuron in one layer is connected with every neuron of the next layer. We developed a multi-channel CNN based on [56], using the *tensorflow.keras* module API. Due to performance reasons, we pre-defined a feature space and systematically searched it using different train/test splits.

### 3.6. BERT

BERT's architecture [57] is based on a multi-layer transformer encoder proposed in [58]. Through extensive pre-training, BERT generates hidden output layers with 768 dimensions for the BERT$_{base}$ model, consisting of embeddings on both the sentence level and the word-level. This model has 12 layers, 12 self-attention heads and 110 million parameters [57]. To tailor BERT for specific tasks, such as classification, fine-tuning can be performed using task-specific training data [59]. For this, an additional output layer must be connected to BERT's hidden output layer. A version of BERT customised for classifying German language texts was developed in [60]. We used this model, adapting it to our specific scenario with four output classes, and fine-tuned it using our Tweets.

### 3.7. Evaluation Metrics

For the evaluation of our models, we employed the traditional metrics of accuracy, precision, recall, and F1 score (where $T$ is true, $F$ is false, $P$ is positive, and $N$ is negative):

$$Accuracy = \frac{TP + TN}{TP + FP + TN + FN} \tag{2}$$

$$Precision = \frac{TP}{TP + FP} \tag{3}$$

$$Recall = \frac{TP}{TP + FN} \tag{4}$$

$$F1\ score = \frac{2 \times TP}{2 \times TP + FP + FN} \tag{5}$$

Additionally, we decided to use a Gaussian scoring function to mirror the order of categories as ordinal data. It presents an additional evaluation metric that shows how far off a misclassification was. A high precision for a class but a relatively low Gaussian score indicates that the misclassifications are very far away from the ground truth, while a low class precision but a high Gaussian score indicates that most of the classifications are grouped in a similar class. Equation (6) shows how the score was calculated:

$$Gaussian\ score = \frac{1}{\sigma\sqrt{2\pi}}e^{-\frac{1}{2}\left(\frac{Ed-\mu}{\sigma}\right)^2} \tag{6}$$

where $Ed$ is $|P - T|$ between the true class $T$ and the predicted class $P$, $\sigma$ is the standard deviation of the distribution and $\mu$ is the mean of the distribution. In our research, $\sigma$ was set to 1.5 and $\mu$ to 0 since these parameters best distinguished our classes.

## 4. Results

In this section, we present the results of the different methods. All algorithms used the same training and validation data sets as well as the same encoding. Algorithm 1 explains how data can be classified by the models, using the BERT model as an example. Table 2 compares the average performance metrics for all categories of each model. Table 3 lists the F1 score and Gaussian score for each model and the respective labelling categories.

---

**Algorithm 1:** Inference by relevance classification model

---

**Input:** Singular Tweets
**if** *input is Pandas DataFrame* **then**
    **if** *length of text > 0 after pre-processing* **then**
        embedding creation by GBERT$_{base}$;
        inference by BERT-based classification model;
        **Output:** four categories:
            *very relevant*
            *rather relevant*
            *barely relevant*
            *not relevant*
    **end**
**end**

---

**Table 2.** Performance comparison of the different algorithms for averaged precision, recall, F1 score and Gaussian score (GS). The respective highest value is shown in bold.

| Model | Accuracy | Precision | Recall | F1 Score | GS |
|-------|----------|-----------|--------|----------|-----|
| NB | 0.40 | 0.40 | 0.40 | 0.40 | 0.70 |
| RF | 0.44 | 0.45 | 0.44 | 0.45 | 0.73 |
| SVM | 0.28 | 0.28 | 0.28 | 0.28 | 0.65 |
| CNN | 0.51 | 0.54 | 0.51 | 0.52 | 0.84 |
| BERT | **0.71** | **0.71** | **0.71** | **0.71** | **0.90** |

**Table 3.** Precision (P), recall (R), F1 score and Gaussian score (GS) for each model and relevance category. The respective highest value is shown in bold.

| | Relevance Categories | | | | | | | |
| --- | --- | --- | --- | --- | --- | --- | --- | --- |
| | 1—Very Relevant | | | | 2— Rather Relevant | | | |
| **Model** | *P* | *R* | *F1* | *GS* | *P* | *R* | *F1* | *GS* |
| NB | 0.39 | 0.32 | 0.35 | 0.58 | 0.38 | 0.47 | 0.42 | 0.74 |
| RF | 0.44 | 0.40 | 0.42 | 0.59 | 0.44 | 0.50 | 0.47 | 0.79 |
| SVM | 0.38 | 0.41 | 0.40 | 0.56 | 0.26 | 0.22 | 0.24 | 0.68 |
| CNN | 0.62 | 0.45 | 0.53 | 0.83 | 0.38 | 0.44 | 0.41 | 0.80 |
| BERT | **0.76** | **0.64** | **0.69** | **0.89** | **0.63** | **0.69** | **0.66** | **0.89** |
| | 3—Barely Relevant | | | | 4—Not Relevant | | | |
| **Model** | *P* | *R* | *F1* | *GS* | *P* | *R* | *F1* | *GS* |
| NB | 0.38 | 0.32 | 0.35 | 0.73 | 0.44 | 0.49 | 0.46 | 0.74 |
| RF | 0.35 | 0.36 | 0.36 | 0.74 | 0.56 | 0.51 | 0.53 | 0.78 |
| SVM | 0.17 | 0.18 | 0.18 | 0.69 | 0.30 | 0.31 | 0.31 | 0.67 |
| CNN | 0.50 | **0.68** | 0.58 | 0.84 | 0.64 | 0.47 | 0.54 | 0.87 |
| BERT | **0.72** | 0.64 | **0.68** | **0.90** | **0.73** | **0.86** | **0.79** | **0.9** |

Since BERT consistently provided the best results, we will focus on its classification results. In terms of recall and precision in particular, there were enormous differences, although all classifiers received the same input data. Figure 3 shows how many Tweets of the validation data set were assigned to their respective relevance class, and the class distribution after assigning the classes using BERT. After classification, there were minor shifts in the balanced, stratified data set. The figure reveals a certain bias of the model for the categories "2—rather relevant" and "4—not relevant", into which slightly more Tweets than desired were placed. Additionally, Figure 4 shows how the output classes were composed. While the majority of Tweets were classified in the correct category for all relevance categories, there were some misclassifications. Tweets from every other category were represented in each class; a few very relevant Tweets, for example, were assessed as irrelevant. The only exception was the category "1—very relevant", which did not receive any irrelevant Tweets. However, the figure also shows that the misclassifications were usually in semantically similar categories. The same can be said for all classifiers in Table 3, where the Gaussian scores were also appropriate for models with very low recall and precision.

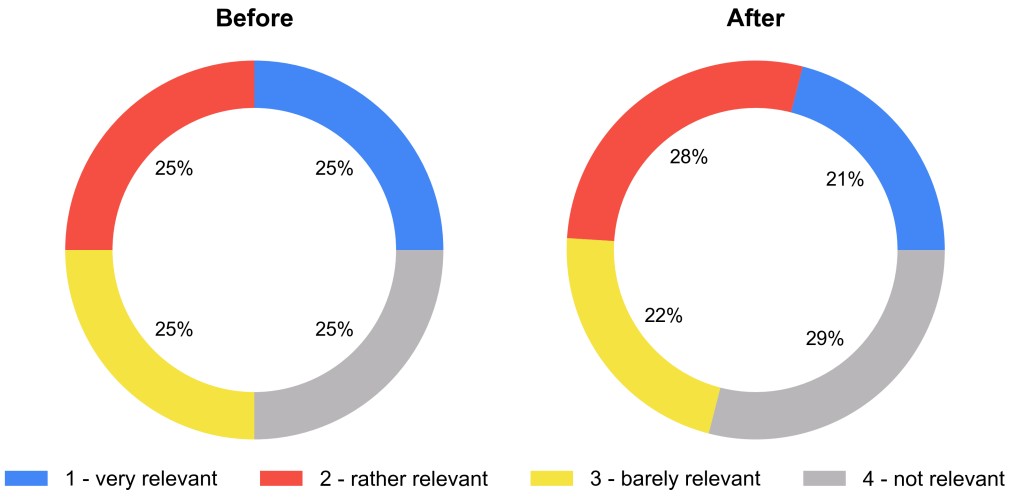

**Figure 3.** Class distribution of relevance classes before and after classification for BERT.

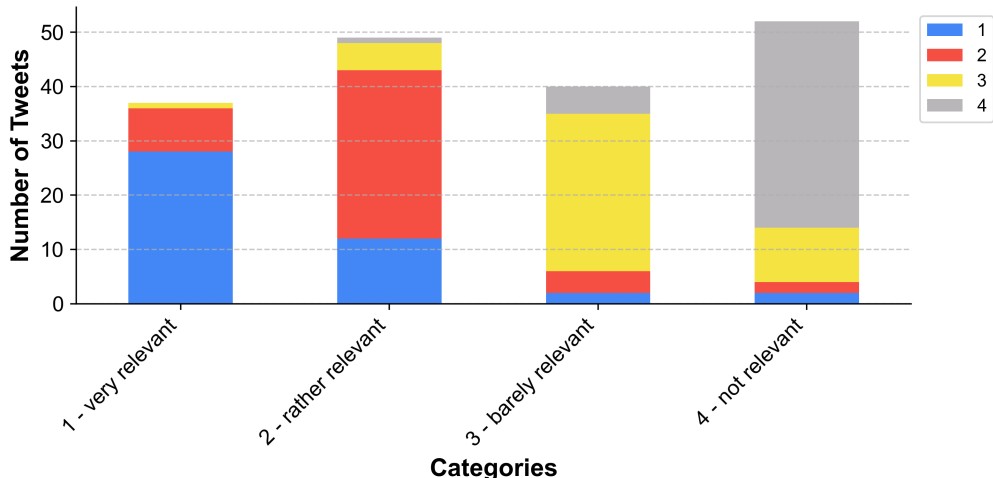

**Figure 4.** Classification composition of BERT.

## 5. Discussion

In the following section, our results are discussed and the applied methods are critically reviewed.

### 5.1. Training Data

The requirements for the Tweets to be written in German and related to a flooding event led to data scarcity. For this reason, the data collection process was carefully developed to maximise the randomness while keeping the quality of the training data high. The direct collaboration with first aid organisations in this process should be emphasised here, which ensured the applicability of the model outputs. Nevertheless, a larger training data set could have potentially improved our results. By providing an extensive labelling guide, we tried to turn the labelling of each Tweet from something highly subjective [11] to something relatively objective and comparable [61]. Using a 2/3 majority for the inter-annotator agreement ensured that the class objectivity of each Tweet was maximised. Yet, this approach only ensured the overall coherence, but not the actual quality, of the labelled data.

In the first step of the training phase, the imbalanced distribution of Tweets in the four classes was balanced. Even though scaling down three of the four classes inevitably led to a loss of information, it tackled one of the major problems, i.e., the drop in classification performance for one class while trying to gain it for another [62], successfully.

### 5.2. Results

The algorithms used in our research can be split into two categories: older methods (NB, RF, SVM) and newer, computationally more extensive methods (CNN, BERT). Comparing those two categories showed that the older algorithms generally scored lower across all used metrics. Overall, the SVM performed the worst. This is not surprising since this method is rather feature engineering extensive and our input data were only tokenised. The results of NB and RF were quite similar. For the newer algorithms, BERT mostly outperformed the CNN. Only for class "3—barely relevant" did the CNN have a 0.04 higher recall.

We proposed an additional Gaussian scoring function for a more precise model evaluation. For the Gaussian score, BERT outperformed the CNN again by 0.06. Interestingly, the CNN matched the Gaussian score of BERT for category "1—very relevant" and even outperformed it by 0.02 for category "3—barely relevant". Since BERT performed better on accuracy, precision, recall and F1 score, this indicates that BERT had fewer misclassifications but they were semantically further away from their assigned label.

Ref. [30] achieved an overall accuracy between 0.50 and 0.67, F1 scores between 0.47 and 0.63 and a recall between 0.45 and 0.63 for their classes with a CNN. The highest macro-F1 score achieved by [31] was 0.66 for their linear SVM model. With their stacked method, Ref. [33] could classify Tweets with a precision, recall and F1 score of 0.76, and [37], who also used a BERT model, obtained an accuracy, precision and F1 score of 0.78 and a recall of 0.79. Accordingly, the evaluation metrics achieved by our BERT model specification for German-language Tweets (accuracy, precision, recall and F1 score of 0.71) correspond to the current state of the art. Additionally, we were also able to show that its misclassifications were mostly in semantically similar classes.

### 5.3. Limitations

Due to the low ratio of potentially relevant Tweets in our data set, it was necessary to increase the likelihood of querying Tweets which are related to a flooding event. However, this also limited our training data size. Additionally, some of the selected keywords are not exclusively used in connection to flooding events. For example, "Höchststand" (peak) was a keyword which was commonly featured in Tweets referring to the COVID-19 pandemic, while "Gewitter" (thunderstorm) and "Sturm" (storm) are terms commonly utilised in weather forecasts. Accordingly, our pre-filtered data set also contained numerous Tweets that fell into category 4.

Furthermore, our methodology merely focused on classifying Tweets based on text information. Hence, we assumed that each classified Tweet was temporally and spatially related to a flood event. However, the explicit inclusion of timestamp and geometry in a relevance classification would be appropriate. We decided against feeding this information to our complex models in order to avoid a bias of the models towards single events (e.g., the Ahr Valley flood). In the future, this information should still be incorporated, e.g., by integrating it into the labelling process or through additional weighting of the outputs.

In our research, we only considered data from Twitter. However, an application of our models to other social media platforms (e.g., Facebook) should be possible. Since the structure of texts across social media platforms differs, the portability of the models still needs be tested explicitly.

It should be noted that the future of Twitter data as a source for research is unfortunately rather uncertain at the moment. Since the takeover of the social media platform by Elon Musk in late 2022, there have been profound upheavals, especially for the API access and the associated data availability for academic purposes. A polarisation of the discourse in terms of content is also conceivable [43].

### 6. Conclusions

This paper compares various machine learning approaches with respect to their suitability for a multi-class classification problem. Our aim was to define the semantic relevance of a Tweet in relation to a flood event for disaster management purposes. This is crucial because social media data can provide added value in disaster management, but filtering out significant content remains complex.

For this, we focused on German-language Tweets, which had not been targeted previously. In creating our training data set, we worked closely with first aid organisations to define which content is actually relevant for the disaster management process and thus to obtain results that can be used in a real-life application. Our proposed methodology aims to provide the responsible people in a crisis team only with posts that are highly relevant in terms of content and thus to condense the wealth of information.

We found that BERT and the CNN considerably outperformed NB, RF and SVM. Based on our evaluation criteria, we concluded that BERT was the most suitable approach to solve the aforementioned multi-class classification problem.

**Author Contributions:** Conceptualisation, E.B., S.S. and B.R.; methodology, E.B., S.S. and B.R.; software, E.B.; validation, E.B.; formal analysis, E.B.; investigation, E.B. and S.S.; resources, E.B.; data curation, E.B. and S.S.; writing—original draft preparation, E.B. and S.S.; writing—review and editing,

S.S. and B.R.; visualisation, E.B. and S.S.; supervision, B.R.; project administration, B.R.; funding acquisition, B.R. All authors have read and agreed to the published version of the manuscript.

**Funding:** This research was funded by the Austrian Research Promotion Agency (FFG) through the project AIFER (Grant Number 879732). This project has also received funding from the European Commission—European Union under HORIZON EUROPE (HORIZON Research and Innovation Actions) under grant agreement 101093003 (HORIZON-CL4-2022-DATA-01-01). The views and opinions expressed are however those of the author(s) only and do not necessarily reflect those of the European Union—European Commission. Neither the European Commission nor the European Union can be held responsible for them.

**Institutional Review Board Statement:** Not applicable.

**Informed Consent Statement:** Not applicable.

**Data Availability Statement:** Data are contained within the article.

**Acknowledgments:** We would like to thank the Bavarian Red Cross, the Austrian Red Cross and the Federal Agency for Technical Relief (THW) for their help in defining relevance and labelling, particularly Jonas Mehrl, Luisa Knoche and Raimund Görtler. We would like to thank Dorian Arifi, Maximilian Ehrhart, Umut Nefta Kanilmaz and Sophia Ress (University of Salzburg) for their help with labelling. Thank you also to Martin Moser (University of Salzburg) for his feedback on our first draft.

**Conflicts of Interest:** The authors declare no conflicts of interest. The funders had no role in the design of the study; in the collection, analyses, or interpretation of data; in the writing of the manuscript; or in the decision to publish the results.

## Abbreviations

| | |
|---|---|
| ANN | Artificial Neural Network |
| API | Application Programming Interface |
| BERT | Bidirectional Encoder Representations from Transformers |
| CAMM | Cross-Attention Multi-Modal |
| CNN | Convolutional Neural Network |
| GNN | Graph Neural Network |
| KNN | k-nearest neighbour |
| LDA | Latent Dirichlet Allocation |
| NB | Naïve Bayes |
| NLP | Natural Language Processing |
| RF | Random Forest |
| SVM | Support Vector Machine |

## Appendix A

**Table A1.** Flood-related keywords.

| German Keyword | Translation |
|---|---|
| Aufräumarbeiten | cleanup work |
| Bergung | salvage |
| Dammbruch | dam breach |
| Dammschäden | damage to dams |
| Dauerregen | continuous rain |
| Deichbruch | levee breach |
| Deichschäden | damages to levees |
| Einsturz | collapse |
| Erdrutsch | landslide |
| Evakuierung | evacuation |

**Table A1.** *Cont.*

| German Keyword | Translation |
|---|---|
| Extremwetterlage | extreme weather situation |
| Freiwillige Helfer | volunteers |
| Geröll | rubble |
| Gewitter | thunderstorm |
| Großeinsatz | major operation |
| Hangrutschung | landslide |
| Hilfsaktion | relief operation |
| Höchststand | peak/peak level |
| Hochwasser | flood |
| Katastrophe | disaster |
| Krisenstab | crisis management team |
| Luftrettung | air rescue |
| Murgang | mudflow |
| Niederschlag | precipitation |
| Notunterkunft | emergency shelter |
| Orkan | hurricane (European windstorm) |
| Pegel | water level/gauge |
| Platzregen | torrential rain |
| Retentionsfläche | retention area |
| Rettungskräfte | rescue forces |
| Sandsäcke | sandbags |
| Schneeschmelze | snow melting |
| Schlammlawine | mudslide |
| Schutt | debris |
| Starkregen | heavy rain |
| Stromausfall | power outage |
| Sturm | storm |
| Sturzflut | flash flood |
| Tornado | tornado |
| Trümmer | ruins |
| Überflutung | flooding |
| Überschwemmung | inundation |
| Unwetter | severe weather |
| Wasserrettung | water rescue |
| Wiederaufbau | reconstruction |
| Zerstörung | destruction |

**Table A2.** Examples from our labelling guide. The original German Tweets have been translated. Emojis were removed. Usernames were replaced by '@user'.

| Category | Translated Tweet | Reasoning |
|---|---|---|
| | Within one day, the flood water has risen so high that the road is no longer passable. The ferry has stopped operating. #rhine #walsum | flooding/high water level |
| 1—very relevant | Despite rising water levels on the Saale and Weißer Elster rivers, there is no danger of flooding in Halle. The Landesbetrieb für Hochwasserschutz (LHW) has not yet issued a flood warning for Halle. | flood warning |
| | Here in Rheinbach too. Traffic is flowing through the main street again, while the mud is being cleared away there at the same time. | affected infrastructure/damage |

**Table A2.** *Cont.*

| Category | Translated Tweet | Reasoning |
|---|---|---|
| 2—rather relevant | We are lucky that our cellar was not flooded. | non-affected people |
| | Watch out #FakeNews Share @user report. #flood disaster #disasterarea #weareVOST #VOST #SMEM | reference to emergency forces |
| 3—barely relevant | I feel very sorry for the people in NRW. Keep your fingers crossed for all of them. The only thing it can be about now is helping. #Floods | declarations of solidarity |
| | @user Seriously? While the rescue measures are still underway and the #FederalPresident flies from Berlin to the Rhineland, they stand in the background and smile? That is disrespectful to the victims and their families and also politically disrespectful... | political or religious statements |
| | Please all join the campaign stop of the @user and concentrate all forces on the essentials and who can, donate! #Flood | fundraising appeals |
| 4—not relevant | I decided to turn up the music excessively loud today, before the neighbour's child, who can only ride a bike if he squeals, starts doing his rounds. | not related to flood event |
| | @user Good morning at now 16.1 °C, overcast/thunderstorm, wind N 2 bft, air pressure 1022 mbar, precipitation risk 26% from 55,599 Siefersheim in Rheinhessen. | |

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
