# Peer review of "Drowning in the Information Flood: Machine-Learning-Based Relevance Classification of Flood-Related Tweets for Disaster Management"

_information, doi:10.3390/info15030149_

Round 1
Reviewer 1 Report
Comments and Suggestions for Authors
The paper compares different machine learning approaches in terms of their suitability for a multiclass classification problem. The paper define the semantic relevance of a Tweet in relation to a flood event for the purposes of disaster management.
Some recommendations regarding the work:
- The testing method of the different automatic learning algorithms is not described in detail
- Pseudocode description of the proposed method is recommended
- The conclusions do not highlight the advantages of the proposed methodology that helps to identify relevant information for a more efficient disaster management.
Reviewer 2 Report
Comments and Suggestions for Authors
The article presented a novel approach for the classification of the flood-related tweets for disaster management, using machine learning techniques. I recommend the acceptance of the manuscript after minor revision:
1. Please check if it is possible to mention the type of disaster (line 1) the first time it is introduced. For example, “natural disaster” in case it refers to floods, earthquakes, and so on.
2. Please check if the title of Figure 1 can be changed, for example “Workflow for flood-related tweets classification” (or a similar formulation). Optionally, please check if it would be worth to include the Training Data and the Testing Data in the figure for a better readability, as other machine learning methodologies from the research literature also use Validation Data.
3. Please check if it is possible to extend Table 1 with a new column in which an example for each category is provided.
4. Please add more details about how the cross validation was performed (please see lines 154-155), which mention that the training data represents 75% and the testing data 25%. Was the data split only once or was it split 4 times (4-fold cross-validation), and then the results were averaged? It is not very clear.
5. Please check if it is possible to compare the results obtained in the current approach using another evaluation metric (for example precision or accuracy), as the comparison presented in lines 299-305 considers the F1Score.
6. Regarding the Appendix (lines 525 – 526), please check if it is possible to also add the English translation of the keywords.
Reviewer 3 Report
Comments and Suggestions for Authors
In this paper, the authors propose a comparison of different machine learning models for classifying the semantic relevance of flood related, German language, tweets. The topic considered by the authors is extremely interesting, although many related approaches have been proposed in the literature on this topic.
In the paper, the authors illustrate an interesting experimental campaign aimed at achieving their goal. Thus, from my point of view, the paper is very strong from the experimental point of view while it does not have a significant theoretical component. However, this aspect, in my opinion, is not very significant since the authors explicitly say from the beginning that their goal is not to define a new approach but to compare a number of classification approaches.
In this respect, however, the paper has much room for improvement in the "Related work" section. Authors should greatly enrich this section by both citing approaches related to semantic classification in social networks and mentioning approaches related to Twitter, even possibly unrelated to semantic classification of tweets. As examples, authors should cite the following papers "Representation, detection and usage of the content semantics of comments in a social platform" and "A framework for investigating the dynamics of user and community sentiments in a social platform," as well as a variety of other approaches related to theirs.
Comments on the Quality of English LanguageThe English is good
Round 2
Reviewer 1 Report
Comments and Suggestions for Authors
The paper was improved with addition implementing details and completed at the conclusions part according to the recommendations made. Consequently, I propose that it be accepted for publication.
Reviewer 3 Report
Comments and Suggestions for Authors
In my opinion, the authors have not made sufficient effort to improve this paper. For me, the weaknesses indicated in my previous review remain.